# Effect of music on preoperative anxiety and postoperative pain in patients undergoing gynecological surgery: A meta-analysis

Li Yang[1‡], Yong Lin [2]*, Yan Long[2]

**1** Luzhou Maternal and Child Health Hospital (Luzhou Second People's Hospital), Anesthesiology Department, Luzhou city, Sichuan Province, China, **2** Luzhou Maternal and Child Health Hospital (Luzhou Second People's Hospital), Gynecology, Luzhou city, Sichuan Province, China.

‡ These authors share first authorship on this work.
* 2043359232@qq.com

## Abstract

### Background

Preoperative anxiety and postoperative pain often afflict patients and may impact surgical safety and postoperative recovery. Therefore, we studied the influence of music for patients with preoperative anxiety and postoperative pain, in order to provide an effective and safe non-pharmacological intervention for patients.

### Method

Studies on music and gynecological operation were searched in PubMed, EMBASE, Web of Science, and the Cochrane Library. NOS and the modified Jadad Scale were used to evaluate the quality of the studies, and the evidence quality was evaluated and graded according to GRADE guidelines. Stata 12.0 software was used for data analysis.

### Results

We compared the difference in preoperative anxiety between the experimental group and the control group. The results showed that anxiety symptoms were significantly relieved in the experimental group that listened to music(WMD = -6.78, 95% CI (-7.54 - -6.03), P = 0.000). This difference was statistically significant, and the quality of evidence was rated as intermediate. We compared the difference in anxiety symptoms before and after listening to music in the experimental group (WMD = -4.39, 95% CI (-5.14 - -3.64), P = 0.000). The results indicated a significant relief in anxiety symptoms after listening to music. The difference was statistically significant, and the quality of evidence was rated as intermediate and low, respectively. We compared the differences in postoperative pain between the experimental group and the control group (WMD = -0.26, 95% CI (-0.43–-0.10), P = 0.002). The results showed that the

**Data availability statement:** All relevant data are within the manuscript

**Funding:** The author(s) received no specific funding for this work.

**Competing interests:** The authors have declared that no competing interests exist.

pain symptoms of the experimental group, who listened to music, were significantly relieved. The difference was statistically significant, and the quality of evidence was rated as intermediate.

## Conclusion

This meta-analysis showed that listening to music significantly improved preoperative anxiety and postoperative pain during gynecological surgery. Therefore, operating room nurses should perform music intervention in patients undergoing gynecological surgery before and after surgery.

## 1. Background

Preoperative anxiety is mainly manifested as worries and fears about the surgery, such as tachycardia, arrhythmia, or hypertension [1,2]. It is understandable for patients to experience mild anxiety before and after surgery, as it can enhance their adaptation to the external environment. However, severe anxiety can often interfere with the process of recovery, and be considered a mental disorder, and a disease [3].

According to reports, the overall prevalence of preoperative anxiety among 14,000 surgical patients worldwide is 48%. During the waiting time before surgery, anxiety symptoms worsen in 80% of surgical patients. Common fears of surgical patients include surgical complications, the duration and degree of disability after surgery, general anesthesia and related loss of control, and waking up during or after surgery and experiencing discomfort and pain [4,5].

This is particularly true for gynecologic surgery, which often results in the partial or total removal of organs associated with fertility and female characteristics. Sexual desire may be impaired [6]. For example, the physical and psychological effects after hysterectomy are significant, including hot flashes, night sweats, urinary incontinence, and urge incontinence [7]. Patients have reported fears of pain, feelings of rejection by their spouses, and sensations of dismemberment [8]. The surgical treatment of gynecologic cancer can have both short- and long-term effects on sexual health, emotional well-being, reproductive function, and overall quality of life [9]. Therefore, it is necessary to study the anxiety and pain of patients before and after gynecological surgery.

Preoperative anxiety may lead to increased surgical risks, higher anesthesia requirements. And It also may lead more postoperative complications, including heightened postoperative pain, nausea, vomiting, and prolonged hospital or postoperative stay [10,11].

In addition, preoperative anxiety symptoms also serve as predictors of reduced satisfaction. And preoperative anxiety symptoms are subjective indicators of treatment efficacy from the patient's viewpoint [12,13]. Particular note is that an increase in preoperative anxiety is also a risk factor for postoperative mortality [14–16].

There are different treatment options for preoperative anxiety, including both pharmacological and non-pharmacological methods. Non-pharmacological interventions

are becoming increasingly popular due to the potential adverse effects of pharmacological treatment, such as respiratory problems, drowsiness, interference with anesthetic medication, and prolonged recovery time. These methods include cognitive-behavioral therapy, music therapy, preoperative preparation videos, aromatherapy, hypnosis, guided imagery relaxation therapy, and massage [17]. In conclusion, music intervention may provide a viable alternative to sedatives and anti-anxiety medications to reduce preoperative anxiety in surgical procedures [18].

In surgical procedures, such as adenocarcinoma surgery in women, interventions like music, aromatherapy, massage, and virtual reality have been shown to reduce preoperative anxiety and postoperative pain. These interventions demonstrate cost-effectiveness, minimal invasiveness, and a low risk of side effects. However, this study does not determine which of these measures is the most effective. Music, in particular, does not require significant manpower or material resources, incurs lower costs, and is therefore recommended as the most favorable intervention [19–21].

However, there is no relevant systematic review or meta-analysis to evaluate whether anxiety symptoms in patients before gynecological surgery and postoperative pain can be improved through music intervention. Therefore, we searched for relevant studies to evaluate whether music intervention can alleviate preoperative anxiety and postoperative pain.

## 2. Materials and methods

### 2.1 Inclusion and exclusion criteria

Inclusion criteria:(1) inclusion of randomized controlled trials (RCTs), observational studies, comparative studies, clinical trials, clinical studies, multicenter studies, and case-control studies; (2) female patients undergoing gynecological surgery; including partial hysterectomy,dilatation and curettage, completion curettage, tubal ligation, total hysterectomy, myomectomy, pelvic reconstructive surgery and so on.(3) all experimenters undergo music intervention before and after surgery; (4) outcome measures include preoperative anxiety and postoperative pain. (5) Literature utilizing the State-Trait Anxiety Inventory Y Form (STAI-Y) to assess anxiety. (6) Literature is limited to English only.

Exclusion criteria: (1) animal experiments, cell studies, reviews, meta-analyses, case reports, or letters; (2) literature that reports on the same data repeatedly; (3) literature with data errors that cannot be used; (4) studies with outcome measures that do not meet inclusion criteria or have significant errors or omissions.

### 2.2 Search strategy

To identify studies that meet the inclusion criteria, computer searches and manual searches were conducted in the PubMed, Embase, Web of Science, and Cochrane Library databases from the date of database establishment to April 2024. The search was limited to English-language publications and was not restricted by the year of publication. Search terms included "music" and "gynecological"or "preoperative anxiety" or "postoperative pain". Two researchers independently conducted computer searches and manual searches of literature, and a third researcher was consulted if there was a disagreement in opinions.

### 2.3 Data extraction

After conducting a literature search, the two researchers (Yong Lin and Yan Long) strictly screened and extracted data from the literature according to the inclusion and exclusion criteria. Finally, each researcher summarized the literature and reached a consensus. The information extracted after screening the literature included: (1) first author and publication date; (2) age and total sample size; (3) study type; (4) intervention measures; (5) outcome measures.

### 2.4 Literature quality assessment

Two researchers independently assessed the quality of the literature using the Newcastle-Ottawa Scale (NOS) and the modified Jadad Scale. Whenever data is missing or additional details are needed, we attempt to collect the necessary

information by contacting the first author. In case of discrepancies during the evaluation process, a consensus was reached by both researchers, or a decision was made by a third researcher.

The Newcastle-Ottawa Scale (NOS) is a tool used to assess the quality of case-control and cohort studies in meta-analysis. It consists of three parts: selection of the study groups, comparability of the study groups, and evaluation of exposure or outcome. A score is given if the study meets relevant criteria. Studies with scores of 0–3, 4–6, and 7–9 are considered to be of low, medium, and high quality, respectively [22].

The modified Jadad Scale is used to evaluate the quality of randomized controlled trials (RCTs). The total score is 7 points. Studies scoring 1–3 points are considered to be of low quality, while those scoring 4–7 points are considered to be of high quality [23].

## 2.5 Statistical analysis

Stata 12.0 was used to conduct a meta-analysis to compare the differences in preoperative anxiety before surgery and the differences in postoperative pain after surgery. The weighted mean difference (WMD) and 95% confidence interval (95% CI) were used to evaluate the data. If the 95% CI did not include 1 or 0, or if $p < 0.05$, then the Weighted Mean Difference (WMD) was considered statistically significant. The heterogeneity among studies was assessed using the Cochran Q and $I2$ statistics, with $P < 0.1$ and $I2 > 50\%$ indicating significant heterogeneity. The funnel plot, as well as the Begg and Egger tests, were used to evaluate publication bias, where $P<0.05$ suggests the presence of publication bias.

## 2.6 Risk of bias and sensitivity analysis

Publication bias was assessed subjectively by observing the symmetry of the funnel plot. If the funnel plot was symmetrical, we subjectively considered that there was no publication bias. Moreover, we also used the Begg test and Egger test to identify publication bias in the literature data, complementing the subjective evaluation. A p-value > 0.05 indicates that the funnel plot was symmetrical, suggesting no publication bias.

Sensitivity analysis was conducted by excluding each study individually and performing subgroup analysis as necessary to investigate the influence of different study characteristics on the observed effects and to identify potential sources of heterogeneity.

## 2.7 Evaluation of the quality of evidence process

Two researchers independently used the GRADE approach to evaluate the certainty of the evidence. For randomized controlled trials, we evaluated the quality of evidence for each outcome based on the risk of bias, inconsistency, indirectness, publication bias, and imprecision. For observational studies, we evaluated the quality of evidence for each outcome based on effect size, dose-response relationship, and negative offset. The evaluation was performed independently by two researchers. In case of disagreements, a third reviewer was consulted to reach a consensus. The quality of evidence was classified into four levels: high, moderate, low, and very low. High-quality evidence was defined as evidence from RCTs (without factors that could cause the quality of evidence to be downgraded) or observational studies that were upgraded in quality by two levels. Moderate-quality evidence is defined as evidence from RCTs that were downgraded by one level or observational studies that were upgraded by one level. Low-quality evidence was defined as evidence from RCTs that were downgraded by two levels or observational studies. Very low-quality evidence is defined as evidence from RCTs that were downgraded by three levels, observational studies that were downgraded by one level, case series, and case reports [24,25].

## 2.8 Ethical approval

All analyses were based on published studies and did not require ethical approval or patient consent.

# 3 Results

## 3.1 Procedure of study retrieval and characteristics of the study

The initial search produced 136 studies. After reviewing titles and abstracts, 18 studies remained. After checking for repeated studies and reading the full text, 5 studies were finally included in the meta-analysis [26–30]. The retrieval process of the studies is shown in Fig 1. The basic characteristics of the included studies are shown in Table 1.

## 3.2 Risk of bias assessment of included studies

The Jadad scale and NOS were used to evaluate the risk of bias in the included studies. The Jadad scale was used to evaluate the quality of randomized controlled trials. The quality scores of the relevant articles ranged from 4 to 5 points. The NOS was used to evaluate the quality of case-control studies, with the quality scores of the relevant studies being 6 points. All studies were of high quality. The quality scores of the studies are shown in Table 2 and 3.

## 3.3 Assessment of the level of evidence quality

### 3.3.1 Meta-analysis comparing preoperative anxiety of patients in the experimental and control groups.
This meta-analysis included three studies. One case-control study had a large effect size, without any significant confounding factors, and the quality of evidence was upgraded by one level. Two randomized controlled trials had a risk of bias, and the quality of evidence was downgraded by one level. Therefore, the level of evidence quality for comparing preoperative anxiety of patients in the experimental and control groups was moderate. The results are shown in S1 Table.

### 3.3.2 Meta-analysis comparing preoperative anxiety before and after patients listen to music.
This meta-analysis included three studies. One study had a large effect size and no significant confounding factors, so the quality of evidence for the study was upgraded by one level. Therefore, the level of evidence quality for comparing preoperative anxiety before and after patients listen to music was moderate and low.The results are shown in S1 Table.

### 3.3.3 Meta-analysis comparing pain in patients who listened to music and control patients after surgery.
This meta-analysis included three studies. Two studies were randomized controlled trials with a risk of bias, so the quality of evidence for each study was downgraded by one level. Another study was a case-control study with a large effect size and no significant confounding factors, so the quality of evidence was upgraded by one level. Therefore, the level of evidence quality for comparing postoperative pain after patients listen to music was moderate. This results of the evidence quality assessment of the studies are shown in S1 Table.

## 3.4 Meta-analysis results

### 3.4.1 Meta-analysis comparing preoperative anxiety of patients in the experimental and control groups.
This meta-analysis included three studies. We compared the difference about preoperative anxiety between in the experimental and control groups in gynecological surgery. The results showed a significant difference in patients in the experimental and control groups (WMD = -6.78, 95% CI (-7.54 - -6.03), P = 0.000) (Begg's Test p=1, Egger's Test p=0.072). This suggests that preoperative anxiety significantly decreased in patients who listened to music before surgery. This result is shown in Fig 2.

### 3.4.2 Meta-analysis comparing preoperative anxiety before and after patients listen to music.
This meta-analysis included three studies. We compared the difference about preoperative anxiety before and after patients listen to music before gynecological surgery. The results showed a significant difference before and after patients listen to music (WMD = -4.39, 95% CI (-5.14 - -3.64), P = 0.000) (Begg's test p=1.000, Egger's test p=0.857). This suggests that anxiety symptoms in patients were significantly alleviated after listening to music.This result is shown in Fig 3.

### 3.4.3 Meta-analysis comparing postoperative pain in patients who Listened to music and control patients after surgery.
This meta-analysis included three studies. We compared the difference in postoperative pain between patients

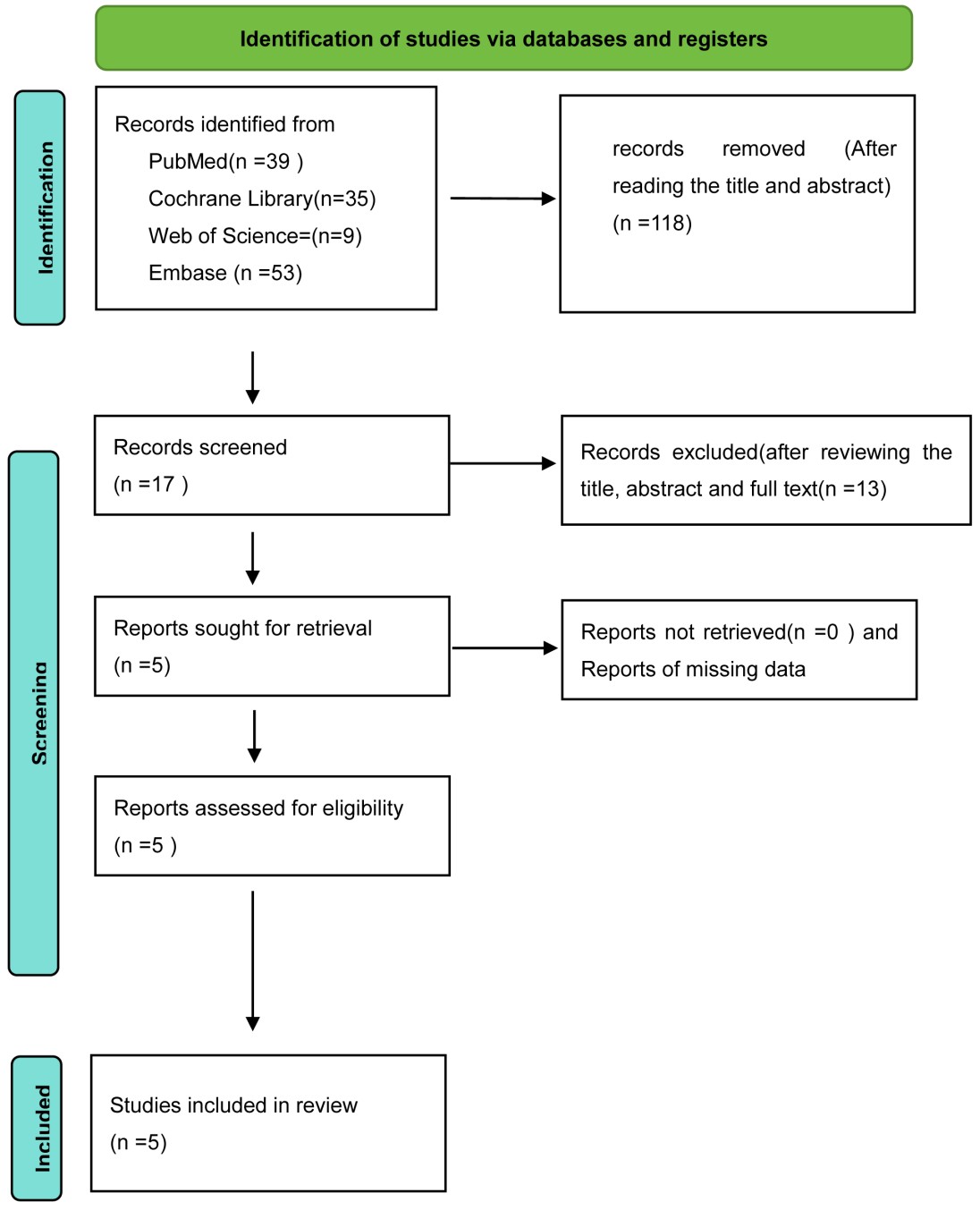

**Fig 1. The retrieval process of the studies.**

who listened to music and control patients after surgery. The results showed a significant difference between patients who listened to music and normal patients after surgery in terms of postoperative pain (WMD = -0.26, 95% CI (-0.43–-0.10), P = 0.002) (Begg's test p=1.000, Egger's test p=0.792). This suggests that patients who listened to music after surgery had significant relief in postoperative pain.This result is shown in Fig 4.

**Table 1. The basic characteristics of the included studies.**

| Study | Study type | Intervening measure | Age (years) | | Sample | Time of collection | Outcome |
|---|---|---|---|---|---|---|---|
| | | | Experimental group | Control group | Experimental group/ Control group | | |
| Leodoro J Labrague 2016 | case-control trial | music (before surgery) | 42.31 ± 4.89 | 42.10 ± 5.02 | 48/49 | before surgery | preoperative anxiety |
| Jvan Casarin 2021 | RCT | music (before and after surgery) | 46.6–49.8 | 49.4–51.7 | 30/70 | Before/ 6 hours after surgery | preoperative anxiety, and postoperative pain |
| Marion Good 2008 | case-control trial | music (after surgery) | 37±7 | 46± 9 | 34/39 | the second day after surgery | postoperative pain |
| Eun Kyung Choi 2023 | RCT | music (after surgery) | 46.4 ± 11.8 | 48.3 ± 13.8 | 41/41 | 36 hours after surgery | postoperative pain |
| Yufan Brandon Chen 2021 | RCT | music (before surgery) | 60.2 ±12.3 | 60.8 ±12.5 | 32/34 | before surgery | preoperative anxiety |

**Table 2. Quality scores of the studies. NOS.**

| Study | I | II | III | IV | V | VI | VII | VIII | Total points |
|---|---|---|---|---|---|---|---|---|---|
| Leodoro J Labrague 2016 | 1 | 0 | 1 | 1 | 1 | 1 | 1 | 0 | 6 |
| Marion Good 2008 | 1 | 0 | 1 | 1 | 1 | 1 | 1 | 0 | 6 |

Note:I Case identif-ication is appropriate. II Repres-entative of cases. III The selection of controls. IV Determ-ination of contrast. V The comparability of cases and controls was considered for the design and statistical analysis. VI Determi-nation of the exposure factors VII The same method was used to determine the exposure factors in the cases and controls. VIII No response rate.

**Table 3. Quality scores of the studies. Jadad score.**

| Study | Generation of the random sequences | Randomization hidden | blind method | Withdrawal and loss of follow-up | total points |
|---|---|---|---|---|---|
| Jvan Casarin 2021 | 2 | 2 | 0 | 0 | 4 |
| Eun Kyung Choi 2023 | 2 | 1 | 2 | 0 | 5 |
| Yufan Brandon Chen 2021 | 2 | 2 | 0 | 1 | 5 |

## 3.5 Sensitivity analysis

A sensitivity analysis was conducted by removing one study to evaluate its impact on the combined results. Using the leave-one-out method, the results showed that none of the individual studies had a significant impact on the meta-analysis results, as indicated by a 95% CI. This indicates that the overall conclusions of the meta-analysis are relatively reliable. The results of the sensitivity analysis are shown in Fig 5.

## 3.6 Publication bias

The funnel plots of the included studies are shown in Fig. All of them exhibit roughly symmetry. In addition, we conducted the Begg's test and the Egger's test to evaluate the presence of publication bias in this study. No significant publication bias was found for any of the results.The results are shown in Figs 6-8.

## 4. Discussion

In this paper, we analyzed the effects of music on preoperative anxiety and postoperative pain in patients in gynecological sur-gery. This meta-analysis indicates that patients who listened to music before surgery had significantly improved preoperative

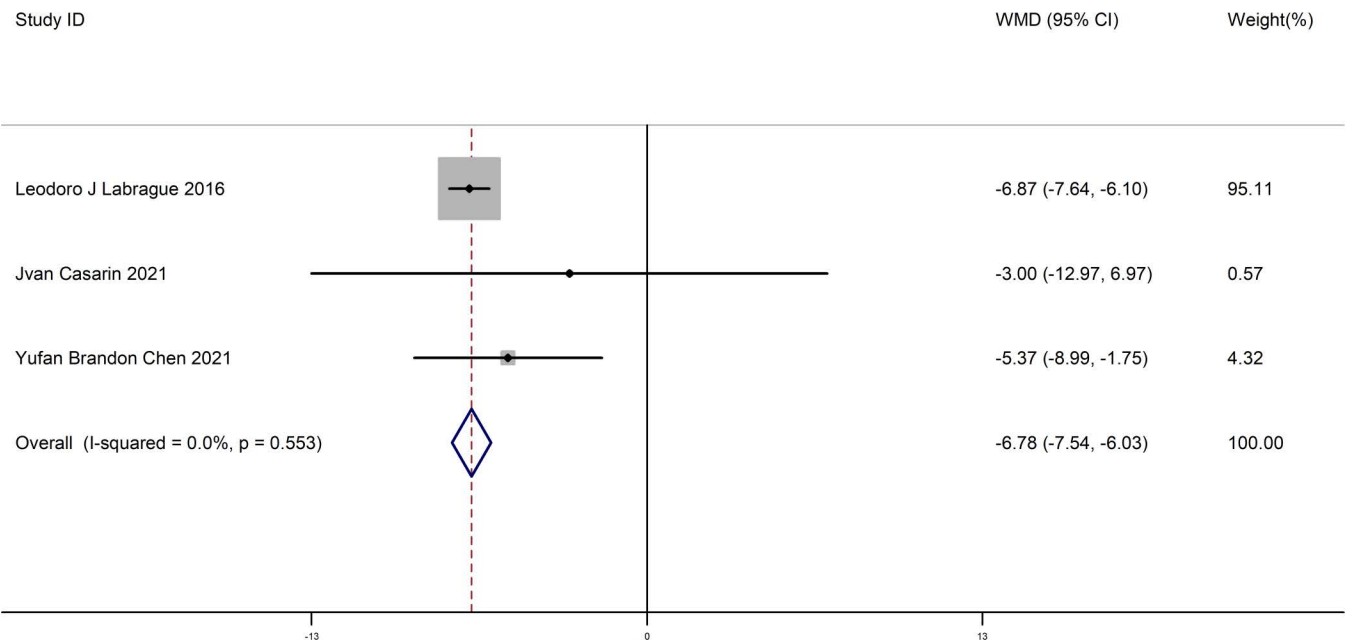

**Fig 2. The forest map comparing preoperative anxiety of patients in the experimental and control groups.**

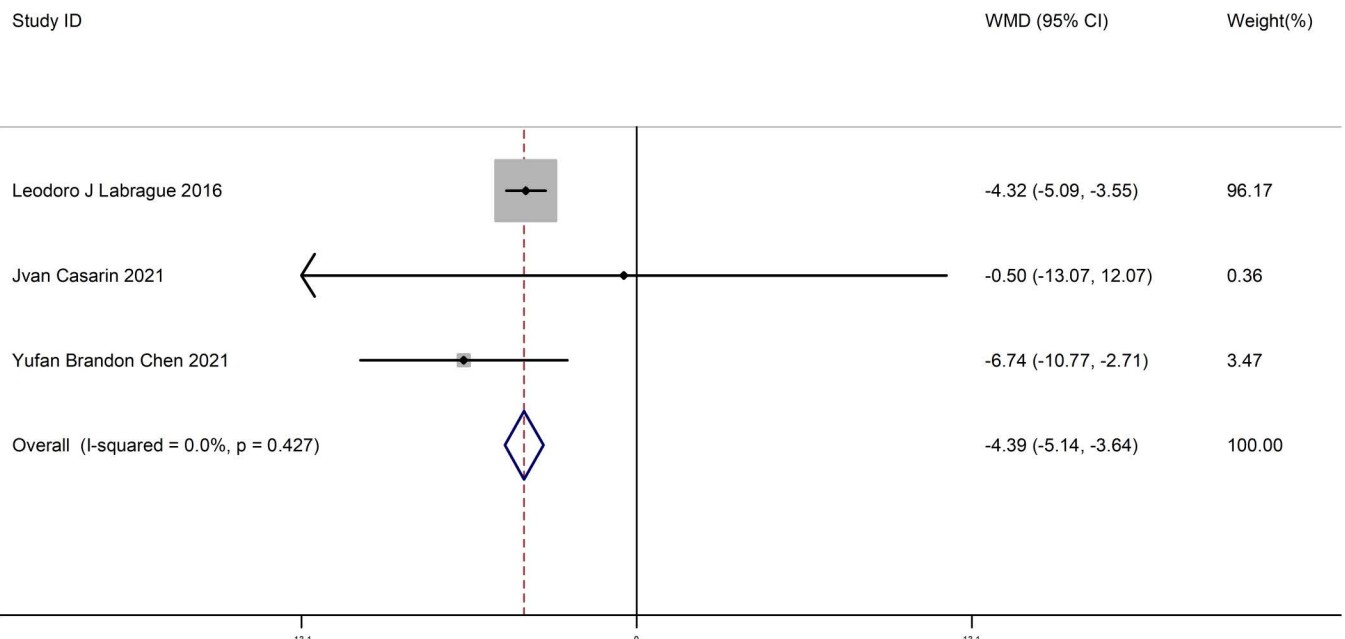

**Fig 3. The forest map comparing preoperative anxiety before and after patients listen to music.**

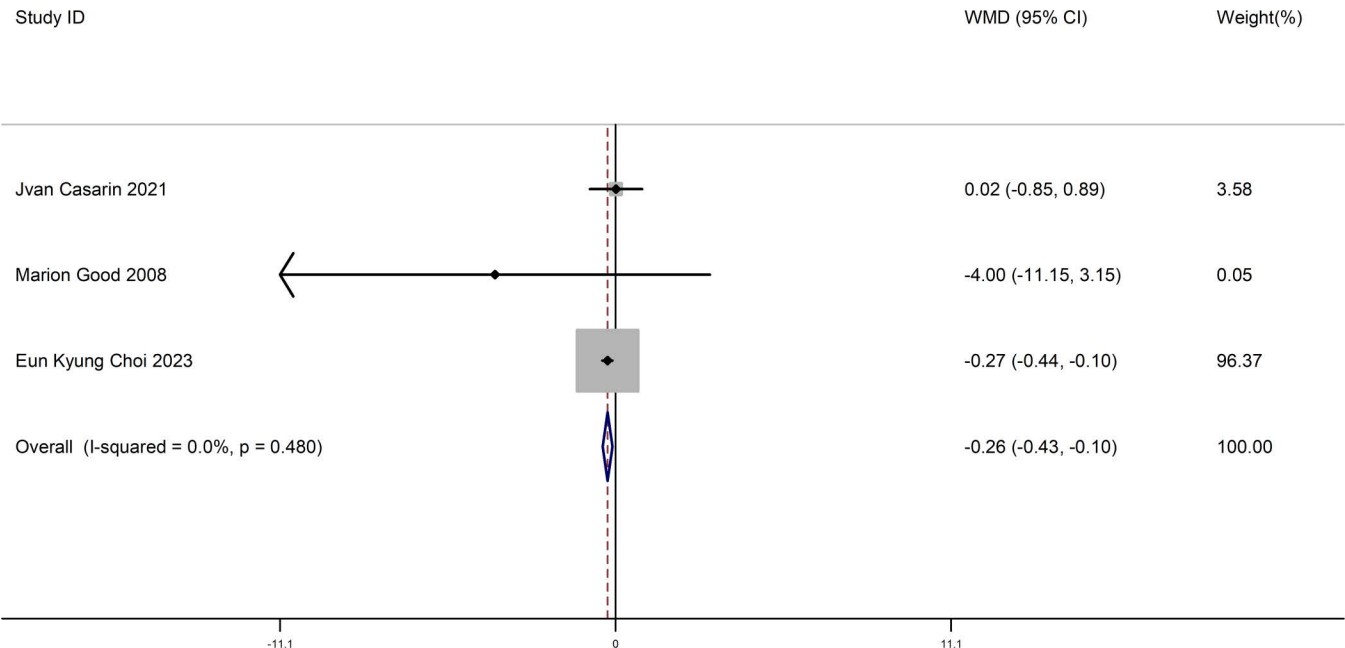

**Fig 4. The forest map comparing postoperative pain in patients who Listened to music and control patients after surgery.**

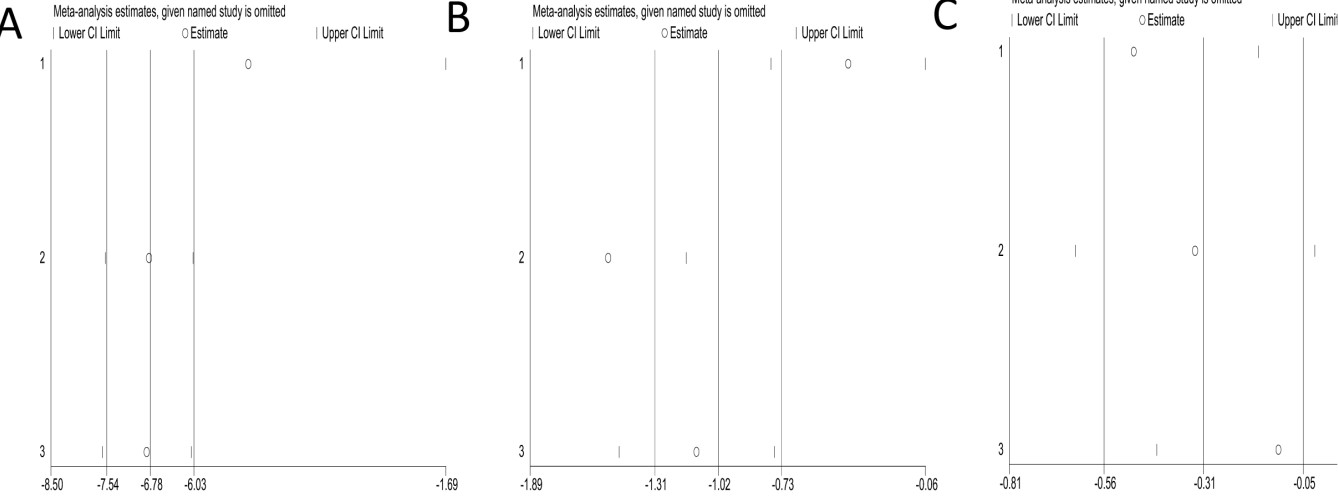

**Fig 5. Sensitivity analysis.** A Meta-analysis comparing preoperative anxiety of patients in the experimental and control groups. B Meta-analysis comparing preoperative anxiety before and after patients listen to music.trimming method: before and after iteration, both fixed effect model and random effect model are meaningful, and the results are robust. C Meta-analysis comparing pain in patients who listened to music and control patients after surger.

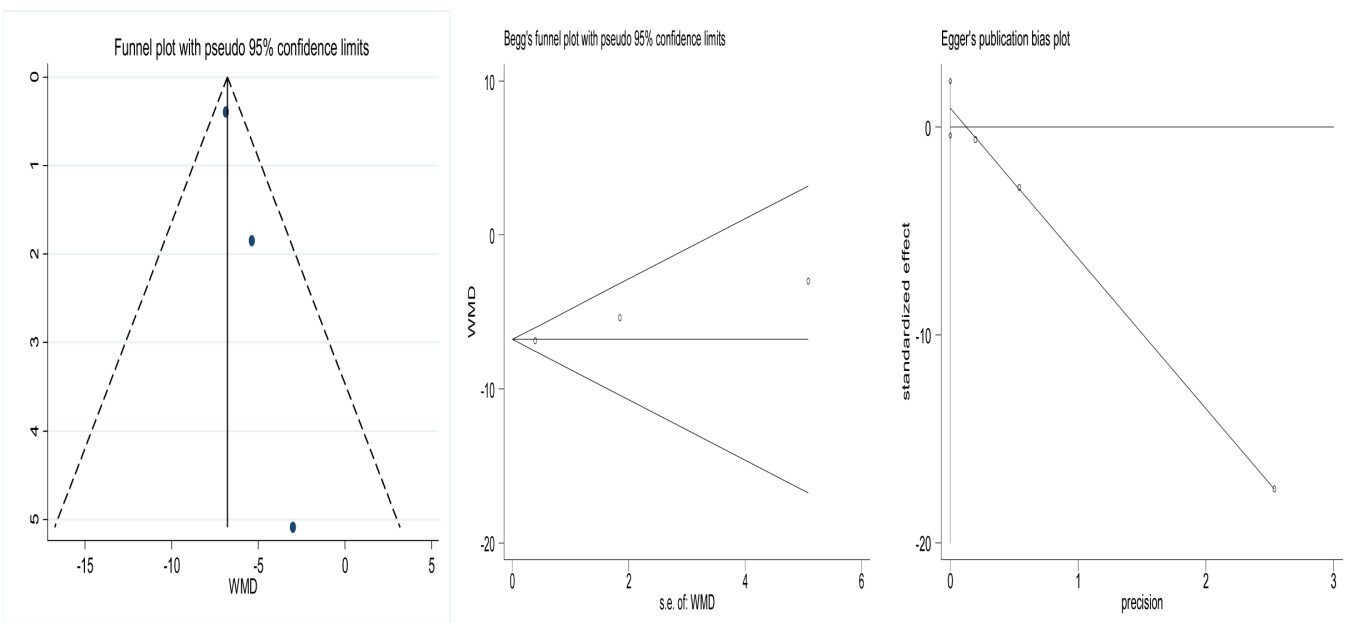

**Fig 6. Funnel plot comparing preoperative anxiety of patients in the experimental and control groups. Begg's Test p =1, Egger's Test p=0.072.**

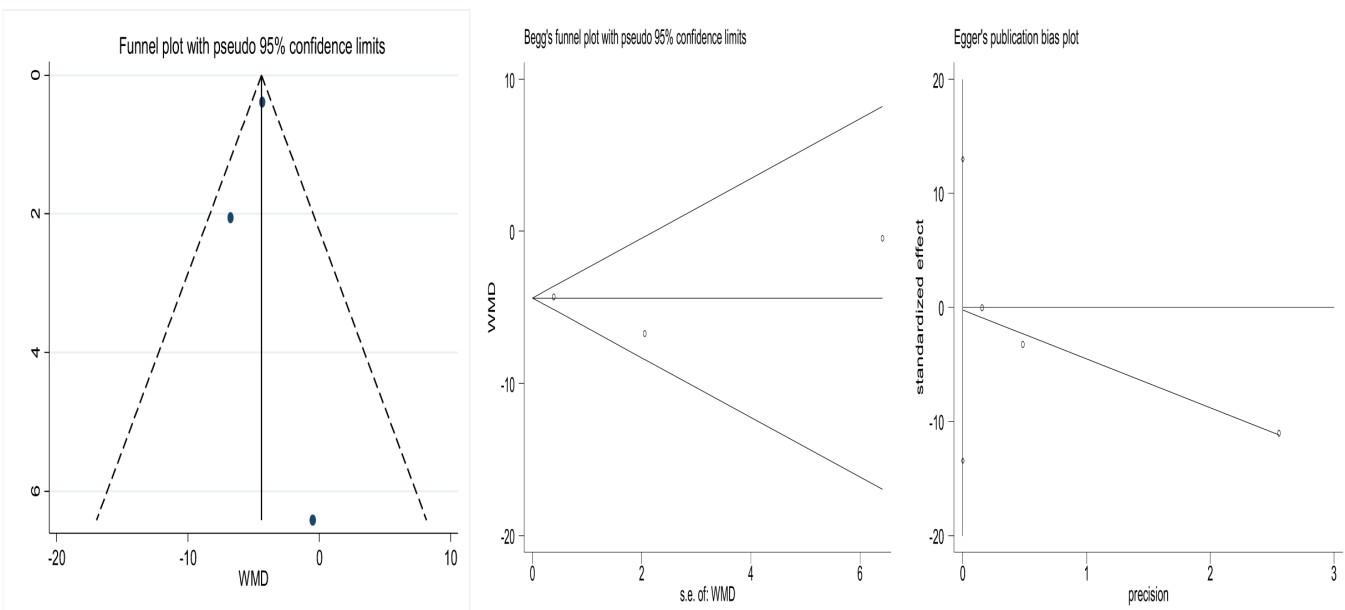

**Fig 7. Funnel plot comparing preoperative anxiety before and after patients listen to music.Begg's test p =1.000, Egger's test p=0.857.**

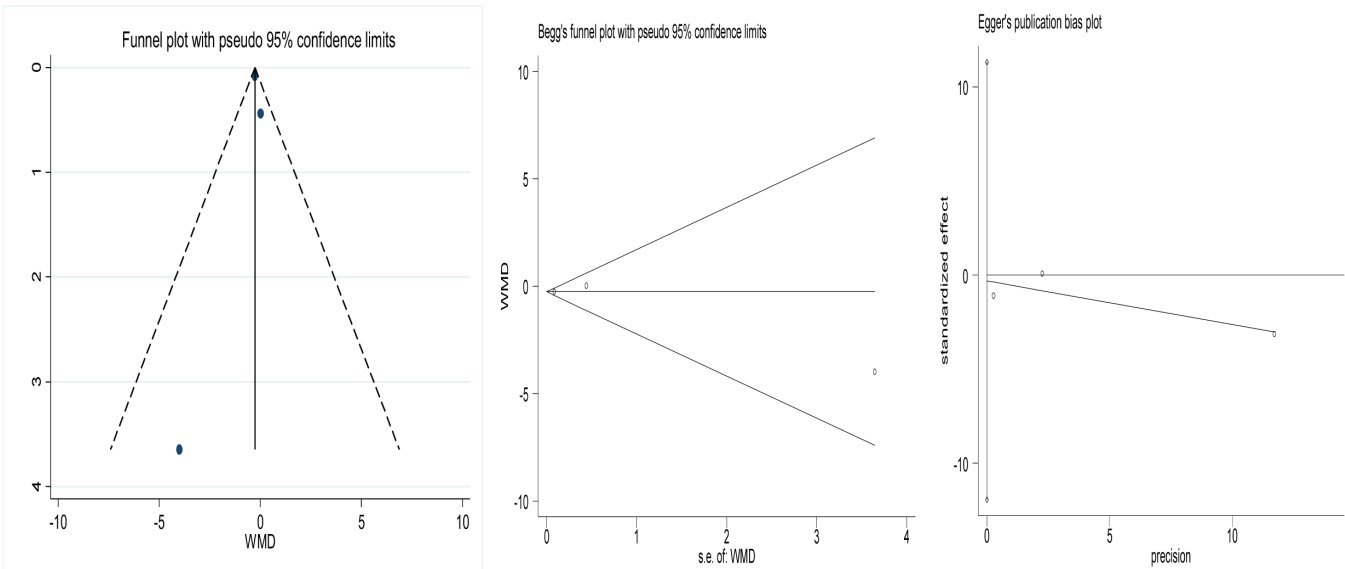

**Fig 8. Funnel plot comparing pain in patients who listened to music and control patients after surgery. Begg's Test p =1, Egger's Test p=0.792.**

anxiety compared to the control group.This meta-analysis also indicates that patients who listened to music after surgery had significant relief in postoperative pain. Moreover, there was a significant improvement in anxiety symptoms after listening to music compared to before. In summary, we should listen to music for patients before and after gynecological surgery.Listening to music can alleviate preoperative anxiety and postoperative pain in patients undergoing gynecological surgery.

Music is one of the most commonly used non-pharmacological interventions to reduce anxiety. Music is a way for people to relax [31]. It helps patients overcome emotional and physical alienation, providing comfort and familiarity in an improved environment, and serves as a pleasant distraction from pain and anxiety [32]. Music significantly reduces preoperative patients' anxiety, heart rate, and other vital signs [33].

We analyzed the mechanism by which music affects patient anxiety.Music is a powerful stimulus that evokes and regulates emotions [34]. Studies have shown that the subject's anxiety may be associated with amygdala activity, and music has a significant impact on the amygdala [35–37]. Music can also have a direct impact on the limbic system of the brain, the reticular structure of the brainstem, and the body's internal organs and functions. For people, harmonious melodies can enhance mental well-being, divert attention, promote relaxation, and reduce tension and anxiety [38].

Regarding the effect of music on anxiety, other studies have also confirmed the findings of this meta-analysis. In a study on anxiety during cesarean section, researchers found that the anxiety levels of patients in the music group during surgery were lower than those of the control group. Another study also showed that,the use of music-based intervention to reduce postoperative opioid requirements, and music-based intervention may also reduce anxiety, depression, and pain catastrophizing [39,40]. It was consistent with the results of our study.

In a study on music intervention for dental anxiety patients, it was found that music significantly reduced heart rate, systolic blood pressure fluctuation, diastolic blood pressure fluctuation, anxiety scores, pain scores. Music also significantly decreased anxiety and pain levels while improving patient cooperation [41]. It was consistent with the results of our study.

In a study on music intervention for cataract surgery patients, researchers found that listening to music during cataract surgery significantly reduced self-reported anxiety levels before, during, and after the procedure. Moreover, it led to a significant decrease in postoperative blood pressure [42].

Another study also indicated that listening to music is associated with a significant reduction in anxiety and pain after cardiac surgery [43]. A meta analysis showed that music therapy can significantly reduce anxiety, pain, HR, SBP, and postoperative opioid use and even improve SpO2 in patients who undergo cardiothoracic surgery. Music therapy has a positive effect on patients after cardiothoracic surgery with few side effects [44]. It was consistent with the results of our study.

Because of the risk of bias in the randomized controlled trials and the inclusion of observational studies in this paper, the quality of evidence was rated as low to medium. This indicates that there is limited confidence in these results, and the actual outcomes may differ from those reported. Therefore, more and better quality randomized controlled trials and adherence to reporting bias control systems are recommended.

The conclusions of these studies are consistent with the findings of this meta-analysis, indicating that music can alleviate preoperative anxiety and postoperative pain in patients. It's totally confirmed that music is an effective, safe, and inexpensive non-pharmacological intervention. It can reduce preoperative anxiety and postoperative pain in patients. This is a good choice for preoperative patients to listen to music before and after surgery. It also plays a supportive role for operating room nurses in patient care.

## 5. Strengths and limitations

A strength of this review is the low heterogeneity among gynecological operations. Furthermore, the State-Trait Anxiety Inventory (STAI) for anxiety and the Visual Analog Scale (VAS) for pain, which are reliable, validated, and easy-to-use assessment tools, were the most commonly utilized instruments in the included studies, aiding in clinical interpretation. This study has some limitations. Firstly, potential biases and confounding factors in the study cannot be completely ruled out in this study.And the number of randomized controlled trials included is limited, and more high-quality RCTs are needed to enhance the quality of the meta-analysis. Second, the sample size of the included studies is limited, and more samples need to be included. Third, although the results of the study are robust, some of the findings are based on small-scale studies, which may be affected by publication bias. Therefore, these findings should be interpreted with caution. Finally, the timing, duration, and type of music intervention are different widely according to the studies, and a meta-analysis comparing different music interventions was not possible due to the limited number of studies.

## 6. Future research requirements

Future research on music interventions in gynecological operations should focus on methodological factors. In order to make definite conclusions, multicenter and randomized studies with larger sample sizes should be conducted. Furthermore, there is a lack of literature on the type of music, the duration, and frequency of interventions. Therefore, there is a need for research to study and report on these factors, which would be more conducive to providing guidance for music into clinical practice. This would be a great complement.Finally, the influence of music on postoperative anxiety among patients following gynecological surgery should be investigated to enhance patient support during the surgical recovery phase.

## 7. Conclusion

This meta-analysis showed that listening to music significantly improved preoperative anxiety and postoperative pain during gynecological surgery. Therefore, operating room nurses should perform music intervention in patients undergoing gynecological surgery before and after surgery.

## Supporting information

**S1 Table. Results of the evidence quality assessment of the studies.**
(XLSX)

**S2 Data sheet. Detailed data of included literature.**
(XLSX)

**S3 Risk assessment. Quality scores of the studies**.
(XLSX)

**S4 Literature. All the literature retrieved**.
(XLSX)

## Author contributions

**Writing – original draft:** Yong Lin, Li Yang.

**Writing – review & editing:** Yong Lin, yan long.

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
