## [Decision Letter · Decision Letter 0]

21 Aug 2024

Dear Dr. Lin,

We look forward to receiving your revised manuscript.

Kind regards,

Ahmed Mustafa Rashid

Academic Editor

PLOS ONE

Journal Requirements:

Reviewers' comments:

Reviewer's Responses to Questions

**Comments to the Author**

1. Is the manuscript technically sound, and do the data support the conclusions?

Reviewer #1: Yes

Reviewer #2: Yes

2. Has the statistical analysis been performed appropriately and rigorously?

Reviewer #1: Yes

Reviewer #2: N/A

3. Have the authors made all data underlying the findings in their manuscript fully available?

Reviewer #1: Yes

Reviewer #2: Yes

4. Is the manuscript presented in an intelligible fashion and written in standard English?

Reviewer #1: No

Reviewer #2: Yes

Reviewer #1: Lin et al. present a very interesting study titled “Effect of Music on preoperative anxiety and postoperative pain in patients undergoing gynecological surgery”. This study shows that musical intervention was effective in reducing anxiety and pain in patients undergoing gynaecological surgery. However, to improve the quality of their manuscript I would recommend the authors make the following edits:

1. Background Section, Page 8: The rationale should focus more specifically on music intervention by highlighting its unique benefits. The authors could do so by citing studies or providing evidence demonstrating the effectiveness of music intervention in reducing preoperative anxiety and improving patient outcomes, especially when compared to alternative non-pharmacological approaches.

2. Background Section, Page 8: The background section should focus more on gynaecological surgery, aligning with the research topic. The authors should highlight how studying musical intervention can impact preoperative anxiety in this patient population specifically as this can significantly enhance the study's relevance. Moreover, the authors could talk about how anxiety is notably prevalent in gynaecological operations and how addressing this anxiety can impact surgical outcomes.

3. Methods Section, Page 9: The inclusion criteria states that studies with “females undergoing gynecological surgery," were included, this is too broad and vague. Please specify the types of gynecological surgeries performed in the included studies. If the studies encompassed a range of surgeries, it would be beneficial to mention these in the baseline characteristics table.

4. Results Section, Page 13: The authors have discussed "preoperative anxiety before and after listening to music" in section 3.3.3, which is supposed to address postoperative pain. Please correct this mistake to ensure accuracy and clarity in the manuscript.

5. Discussion Section, Pages 20-21: The discussion section should explicitly address the medical importance and practical implications of the study's results. It is essential to explain exactly how these findings can enhance patient care in clinical settings, especially in gynaecological surgery.

6. Discussion Section, Page 20: The authors have provided a very brief explanation of how music alleviates anxiety and pain, which is insufficient and lacks clarity. Therefore, I would recommend that they expand on this topic by offering a more thorough, specific, and clear explanation of the mechanisms through which music influences anxiety and pain. This could involve providing specific explanations of the physiological processes or other mechanisms supported by the current literature.

7. There are several grammar and typographical issues in the manuscript, that the authors need to address. For instance, in the discussion section, paragraphs vary in spacing, with some double-spaced and others lacking proper spacing. Additionally, different fonts are used sporadically, and there is inconsistent spacing between paragraphs and sections throughout the manuscript. The presentation and formatting of the tables need improvement as insufficient spacing between rows and columns makes it challenging to discern headings and sections clearly. Moreover, the reference list contains unnumbered references, which affects the overall organization and readability of citations.

Reviewer #2: 1. The introduction should be divided into three parts and provide a strong rationale. Additionally, the flow should be improved for better consistency.

2. The inclusion and exclusion criteria paragraph should be mentioned after the search strategy paragraph.

3. Ethical considerations(2.8) should be addressed in the declarations section following the discussion, rather than in the methodology. If they are already included in the declarations section, they should not be repeated.

4. Review the use of abbreviations throughout your article to ensure consistency and avoid repetition.

5. The meta-analysis should be conducted according to PRISMA guidelines, but this study has not mentioned adherence to these guidelines.

6. The author should provide a detailed search strategy as a supplementary file.

7. Headings of paragraphs should be in bold letters, and full words should be used, such as "discussion" instead of "discuss."

8. The fonts are not consistent throughout the manuscript, and there are irregular spaces between the lines which shows lack of professionalism.

9. The discussion section is unsatisfactory; the flow of paragraphs lacks proper continuity. The discussion should begin by stating the findings, followed by an explanation of various studies.

10. Abbreviations should not be placed at the end of the manuscript. Essential abbreviations should be defined at their first mention in the text.

11. The results of the evidence quality assessment of the studies should not be included in the main manuscript text. Instead, they should be provided as supplementary materials, along with the original images from the ROB tool.

**Do you want your identity to be public for this peer review?** For information about this choice, including consent withdrawal, please see our Privacy Policy

Reviewer #1: No

Reviewer #2: **Yes: ** Akash kumar

---

## [Author Response · Author response to Decision Letter 1]

26 Aug 2024

Effect of music on preoperative anxiety and postoperative pain in patients undergoing gynecological surgery: a meta-analysis

Li Yang1* ,Yong Lin1,2 ,Yang Long1

1 LUZHOU MATERNAL AND CHILD HEALTH HOSPITAL (LUZHOU SECOND PEOPLE'S HOSPITAL), Luzhou city, Sichuan Province, China.

* First author

2 Corresponding author

E-mail address: 2043359232@qq.com(LY)

Abstract

Background

Preoperative anxiety and postoperative pain often afflict patients and may impact surgical safety and postoperative recovery. Therefore, we studied the influence of music on patients with preoperative anxiety and postoperative pain, in order to provide an effective and safe non-pharmacological intervention for patients.

Method

Studies on music and gynecological operation were searched in PubMed, EMBASE, Web of Science, and the Cochrane Library. NOS and the modified Jadad Scale were used to evaluate the quality of the studies, and the evidence quality was evaluated and graded according to GRADE guidelines. Stata 12.0 software was used for data analysis.

Results

We compared the difference in preoperative anxiety between the experimental group and the control group. The results showed that anxiety symptoms were significantly relieved in the experimental group that listened to music(WMD = -6.78, 95% CI (-7.54 - -6.03), P = 0.000) . This difference was statistically significant, and the quality of evidence was rated as intermediate. We compared the difference in anxiety symptoms before and after listening to music in the experimental group (WMD = -4.39, 95% CI (-5.14 - -3.64), P = 0.000) . The results indicated a significant relief in anxiety symptoms after listening to music. The difference was statistically significant, and the quality of evidence was rated as intermediate and low, respectively. We compared the differences in postoperative pain between the experimental group and the control group (WMD = -0.26, 95% CI (-0.43–-0.10), P = 0.002). The results showed that the pain symptoms of the experimental group, who listened to music, were significantly relieved. The difference was statistically significant, and the quality of evidence was rated as intermediate.

Conclusion

This meta-analysis showed that listening to music significantly improved preoperative anxiety and postoperative pain during gynecological surgery. Therefore, operating room nurses should perform music intervention in patients undergoing gynecological surgery before and after surgery.

Key words: music, preoperative anxiety, postoperative pain, gynecological surgery, nursing

Background

Preoperative anxiety is mainly manifested as worries and fears about the surgery, such as tachycardia, arrhythmia, or hypertension. [1,2]It is understandable for patients to experience mild anxiety before and after surgery, as it can enhance their adaptation to the external environment. However, severe anxiety can often interfere with the process of recovery, and be considered a mental disorder, and a disease. [3]

According to reports, the overall prevalence of preoperative anxiety among 14,000 surgical patients worldwide is 48%. During the waiting time before surgery, anxiety symptoms worsen in 80% of surgical patients. Common fears of surgical patients include surgical complications, the duration and degree of disability after surgery, general anesthesia and related loss of control, and waking up during or after surgery and experiencing discomfort and pain. [4, 5]

This is particularly true for gynecologic surgery, which often results in the partial or total removal of organs associated with fertility and female characteristics. Sexual desire may be impaired [6]. For example, the physical and psychological effects after hysterectomy are significant, including hot flashes, night sweats, urinary incontinence, and urge incontinence [7]. Patients have reported fears of pain, feelings of rejection by their spouses, and sensations of dismemberment [8]. The surgical treatment of gynecologic cancer can have both short- and long-term effects on sexual health, emotional well-being, reproductive function, and overall quality of life [9].

Preoperative anxiety may lead to increased surgical risks, higher anesthesia requirements, and more postoperative complications, including heightened postoperative pain, nausea, vomiting, and prolonged hospital or postoperative stay. [10, 11]

In addition, preoperative anxiety symptoms also serve as predictors of reduced satisfaction and are subjective indicators of treatment efficacy from the patient's viewpoint. [12,13] Particular note is that an increase in preoperative anxiety is also a risk factor for postoperative mortality. [14-16]

There are different treatment options for preoperative anxiety, including both pharmacological and non-pharmacological methods. Non-pharmacological interventions are becoming increasingly popular due to the potential adverse effects of pharmacological treatment, such as respiratory problems, drowsiness, interference with anesthetic medication, and prolonged recovery time. These methods include cognitive-behavioral therapy, music therapy, preoperative preparation videos, aromatherapy, hypnosis, guided imagery relaxation therapy, and massage. [17]In conclusion, music intervention may provide a viable alternative to sedatives and anti-anxiety medications to reduce preoperative anxiety in surgical procedures. [18]

In surgical procedures, such as adenocarcinoma surgery in women, interventions like music, aromatherapy, massage, and virtual reality have been shown to reduce preoperative anxiety and postoperative pain. These interventions demonstrate cost-effectiveness, minimal invasiveness, and a low risk of side effects. However, this study does not determine which of these measures is the most effective. Music, in particular, does not require significant manpower or material resources, incurs lower costs, and is therefore recommended as the most favorable intervention. [19-21]

However, there is no relevant systematic review or meta-analysis to evaluate whether anxiety symptoms in patients before gynecological surgery can be improved through music intervention. Therefore, we searched for relevant studies to evaluate whether music intervention can alleviate preoperative anxiety and postoperative pain.

2. Materials and methods

2.1 Inclusion and Exclusion Criteria

Inclusion criteria:(1) inclusion of randomized controlled trials (RCTs), observational studies, comparative studies, clinical trials, clinical studies, multicenter studies, and case-control studies; (2) female patients undergoing gynecological surgery; including partial hysterectomy,dilatation and curettage, completion curettage, tubal ligation, total hysterectomy, myomectomy, pelvic reconstructive surgery and so on.(3) all experimenters undergo music intervention before and after surgery; (4) outcome measures include preoperative anxiety and postoperative pain. (5) Literature utilizing the State-Trait Anxiety Inventory Y Form (STAI-Y) to assess anxiety. (6) Literature is limited to English only.

Exclusion criteria: (1) animal experiments, cell studies, reviews, meta-analyses, case reports, or letters; (2) literature that reports on the same data repeatedly; (3) literature with data errors that cannot be used; (4) studies with outcome measures that do not meet inclusion criteria or have significant errors or omissions.

2.2 Search strategy

To identify studies that meet the inclusion criteria, computer searches and manual searches were conducted in the PubMed, Embase, Web of Science, and Cochrane Library databases from the date of database establishment to April 2024. The search was limited to English-language publications and was not restricted by the year of publication. Search terms included "music" and "gynecological”or “preoperative anxiety” or “postoperative pain”. Two researchers independently conducted computer searches and manual searches of literature, and a third researcher was consulted if there was a disagreement in opinions.

2.3 Data extraction

After conducting a literature search, the two researchers (Yong Lin and Yan Long) strictly screened and extracted data from the literature according to the inclusion and exclusion criteria. Finally, each researcher summarized the literature and reached a consensus. The information extracted after screening the literature included: (1) first author and publication date; (2) age and total sample size; (3) study type; (4) intervention measures; (5) outcome measures.

2.4 Literature quality assessment

Two researchers independently assessed the quality of the literature using the Newcastle-Ottawa Scale (NOS) and the modified Jadad Scale. Whenever data is missing or additional details are needed, we attempt to collect the necessary information by contacting the first author. In case of discrepancies during the evaluation process, a consensus was reached by both researchers, or a decision was made by a third researcher.

The Newcastle-Ottawa Scale (NOS) is a tool used to assess the quality of case-control and cohort studies in meta-analysis. It consists of three parts: selection of the study groups, comparability of the study groups, and evaluation of exposure or outcome. A score is given if the study meets relevant criteria. Studies with scores of 0-3, 4-6, and 7-9 are considered to be of low, medium, and high quality, respectively. [22]

The modified Jadad Scale is used to evaluate the quality of randomized controlled trials (RCTs). The total score is 7 points. Studies scoring 1-3 points are considered to be of low quality, while those scoring 4-7 points are considered to be of high quality. [23]

2.5 Statistical analysis

Stata 12.0 was used to conduct a meta-analysis to compare the differences in preoperative anxiety before surgery and the differences in postoperative pain after surgery. The weighted mean difference (WMD) and 95% confidence interval (95% CI) were used to evaluate the data. If the 95% CI did not include 1 or 0, or if p < 0.05, then the Weighted Mean Difference (WMD) was considered statistically significant. The heterogeneity among studies was assessed using the Cochran Q and I2 statistics, with P < 0.1 and I2 > 50% indicating significant heterogeneity. The funnel plot, as well as the Begg and Egger tests, were used to evaluate publication bias, where P<0.05 suggests the presence of publication bias.

2.6 Risk of bias and sensitivity analysis

Publication bias was assessed subjectively by observing the symmetry of the funnel plot. If the funnel plot was symmetrical, we subjectively considered that there was no publication bias. Moreover, we also used the Begg test and Egger test to identify publication bias in the literature data, complementing the subjective evaluation. A p-value > 0.05 indicates that the funnel plot was symmetrical, suggesting no publication bias.

Sensitivity analysis was conducted by excluding each study individually and performing subgroup analysis as necessary to investigate the influence of different study characteristics on the observed effects and to identify potential sources of heterogeneity.

2.7 Evaluation of the quality of evidence process

Two researchers independently used the GRADE approach to evaluate the certainty of the evidence. For randomized controlled trials, we evaluated the quality of evidence for each outcome based on the risk of bias, inconsistency, indirectness, publication bias, and imprecision. For observational studies, we evaluated the quality of evidence for each outcome based on effect size, dose-response relationship, and negative offset. The evaluation was performed independently by two researchers. In case of disagreements, a third reviewer was consulted to reach a consensus. The quality of evidence was classified into four levels: high, moderate, low, and very low. High-quality evidence was defined as evidence from RCTs (without factors that could cause the quality of evidence to be downgraded) or observational studies that were upgraded in quality by two levels. Moderate-quality evidence is defined as evidence from RCTs that were downgraded by one level or observational studies that were upgraded by one level. Low-quality evidence was defined as evidence from RCTs that were downgraded by two levels or observational studies. Very low-quality evidence is defined as evidence from RCTs that were downgraded by three levels, observational studies that were downgraded by one level, case series, and case reports.[24, 25]

3 Results

3.1 Procedure of study retrieval and characteristics of the study

The initial search produced 136 studies. After reviewing titles and abstracts, 18 studies remained. After checking for repeated studies and reading the full text, 5 studies were finally included in the meta-analysis [26-30]. The retrieval process of the studies is shown in Figure 1. The basic characteristics of the included studies are shown in Table 1.

Figure 1 the retrieval process of the studies

Study Study type Intervening measure Age (years) sample Time of

collection

Outcome

Experime-ntal group Control group Experime-ntal group/Co-ntrol group

Leodoro J

Labrague2016 case-con-trol trial music (before surgery) 42.31 ±

4.89 42.10 ±

5.02 48/49 before surgery preoperative anxiety

Jvan Casarin 2021 RCT music (before and after

surgery) 46.6−49.8 49.4−51.7 30/70 Before / 6

hours after

surgery preoperative anxiety, and

postoperative pain

Marion Good 2008 case-con-trol trial music (after surgery) 37±7 46± 9 34/39 the second

day after

surgery postoperative pain

Eun Kyung Choi 2023 RCT music (after surgery) 46.4 ± 11.8 48.3 ± 13.8 41/41 36 hours

after surgery postoperative pain

Yufan Brandon Chen 2021 RCT music (before surgery) 60.2 ±12.3 60.8 ±12.5 32/34 before surgery preoperative anxiety

Table 1 the basic characteristics of the included studies

3.2 Risk of bias assessment of included studies

The Jadad scale and NOS were used to evaluate the risk of bias in the included studies. The Jadad scale was used to evaluate the quality of randomized controlled trials. The quality scores of the relevant articles ranged from 4 to 5 points. The NOS was used to evaluate the quality of case-control studies, with the quality scores of the relevant studies being 6 points. All studies were of high quality. The quality scores of the studies are shown in Table 2.

Table 2A NOS

study Case identif-ication is appropriate Repres-entative of cases The selection

of controls Determ-ination of contrast The comparability

of cases and

controls was considered for

the design

and statistical analysis Determi-nation of the exposure factors The same

method was

used to

determine

the exposure

factors in

the cases

and controls No response rate

Total points

Leodoro J Labrague 2016 1 0 1 1 1 1 1 0 6

Marion Good 2008 1 0 1 1 1 1 1 0 6

Table 2B Jadad score

Study Generation of the random sequences Randomization hidden blind method Withdrawal and loss of follow-up total points

Jvan Casarin 2021 2 2 0 0 4

Eun Kyung Choi 2023 2 1 2 0 5

Yufan Brandon Chen 2021 2 2 0 1 5

Table 2 Quality scores of the studies

3.3 Assessment of the level of evidence quality

3.3.1 Meta-analysis comparing preoperative anxiety of patients in the experimental and control groups

This meta-analysis included three studies. One case-control study had a large effect size, without any significant confounding factors, and the quality of evidence was upgraded by one level. Two randomized controlled trials had a risk of bias, and the quality of evidence was downgraded by one level. Therefore, the level of evidence quality for comparing preoperative anxiety of patients in the experimental and control groups was moderate. The results are shown in S1 Table.

3.3.2 Meta-analysis comparing preoperative anxiety before and after patients listen to music

This meta-analysis included three studies. One study had a large effect size and no significant confounding factors, so the quality of evidence for the study was upgraded by one level. Therefore, the level of evidence quality for comparing preoperative anxiety before and after patients listen to music was moderate and low.The results are shown in S1 Table.

3.3.3 Meta-analysis compari

---

## [Decision Letter · Decision Letter 1]

6 Oct 2024

Dear Dr. Lin,

Thank you for submitting your manuscript to PLOS ONE. After careful consideration, we feel that it has merit but does not fully meet PLOS ONE’s publication criteria as it currently stands. Therefore, we invite you to submit a revised version of the manuscript that addresses the points raised during the review process.

We look forward to receiving your revised manuscript.

Kind regards,

Ahmed Mustafa Rashid

Academic Editor

PLOS ONE

Journal Requirements:

Reviewers' comments:

Reviewer's Responses to Questions

**Comments to the Author**

Reviewer #1: (No Response)

2. Is the manuscript technically sound, and do the data support the conclusions?

Reviewer #1: Yes

3. Has the statistical analysis been performed appropriately and rigorously?

Reviewer #1: Yes

4. Have the authors made all data underlying the findings in their manuscript fully available?

Reviewer #1: Yes

5. Is the manuscript presented in an intelligible fashion and written in standard English?

Reviewer #1: No

Reviewer #1: The authors have addressed the majority of the comments. However, my last comment regarding the formatting and presentation of the tables has not been fully addressed. The tables are still difficult to read due to issues with their formatting, spacing, and overall presentation. I recommend that the authors revise the tables to improve clarity and ensure that the data is easily interpretable.

**Do you want your identity to be public for this peer review?** For information about this choice, including consent withdrawal, please see our Privacy Policy

Reviewer #1: No

---

## [Author Response · Author response to Decision Letter 2]

9 Oct 2024

It has been modified, please check it out in Response to Reviewers.docx

---

## [Decision Letter · Decision Letter 2]

28 Nov 2024

Dear Dr. Lin,

Thank you for submitting your manuscript to PLOS ONE. After careful consideration, we feel that it has merit but does not fully meet PLOS ONE’s publication criteria as it currently stands. Therefore, we invite you to submit a revised version of the manuscript that addresses the points raised during the review process.

We look forward to receiving your revised manuscript.

Kind regards,

Ahmed Mustafa Rashid

Academic Editor

PLOS ONE

Journal Requirements:

Reviewers' comments:

Reviewer's Responses to Questions

**Comments to the Author**

Reviewer #2: All comments have been addressed

Reviewer #3: All comments have been addressed

2. Is the manuscript technically sound, and do the data support the conclusions?

Reviewer #2: Yes

Reviewer #3: Yes

3. Has the statistical analysis been performed appropriately and rigorously?

Reviewer #2: Yes

Reviewer #3: Yes

4. Have the authors made all data underlying the findings in their manuscript fully available?

Reviewer #2: Yes

Reviewer #3: Yes

5. Is the manuscript presented in an intelligible fashion and written in standard English?

Reviewer #2: Yes

Reviewer #3: Yes

Reviewer #2: Paraphrase grammar errors and proofread your manuscript. Carefully review your text for errors in spelling, grammar and consistency or sentence flow.

Reviewer #3: The introduction gives a good overview of preoperative anxiety and how it affects patients, especially in gynecologic surgery. However, it would help to mention the main goal of your study earlier. This way, readers know what to expect right from the start.

In introduction, Some sentences are long and can be broken down into shorter ones for better readability. Try to make your points clear and concise.

Proofread your manuscript.

Discussion should be improved and more explanation and more explanation of current literature should be added. Current discussion is very short.

**Do you want your identity to be public for this peer review?** For information about this choice, including consent withdrawal, please see our Privacy Policy

Reviewer #2: **Yes: ** Akash Kumar

Reviewer #3: **Yes: ** Ahmed Kamal Siddiqi

---

## [Decision Letter · Decision Letter 3]

6 Feb 2025

Effect of music on preoperative anxiety and postoperative pain in patients undergoing gynecological surgery: a meta-analysis

PONE-D-24-22771R3

Dear Dr. Lin,

We’re pleased to inform you that your manuscript has been judged scientifically suitable for publication and will be formally accepted for publication once it meets all outstanding technical requirements.

Kind regards,

Ahmed Mustafa Rashid

Academic Editor

PLOS ONE

Additional Editor Comments (optional):

Reviewers' comments:

Reviewer's Responses to Questions

**Comments to the Author**

Reviewer #2: All comments have been addressed

Reviewer #3: All comments have been addressed

2. Is the manuscript technically sound, and do the data support the conclusions?

Reviewer #2: Yes

Reviewer #3: Yes

3. Has the statistical analysis been performed appropriately and rigorously?

Reviewer #2: Yes

Reviewer #3: Yes

4. Have the authors made all data underlying the findings in their manuscript fully available?

Reviewer #2: Yes

Reviewer #3: Yes

5. Is the manuscript presented in an intelligible fashion and written in standard English?

Reviewer #2: Yes

Reviewer #3: Yes

Reviewer #2: This manuscript does a great job of showing how music can help reduce preoperative anxiety and postoperative pain in patients undergoing gynecological surgery. By using well-established evaluation tools, the study provides reliable and practical evidence. The findings highlight music as an easy, non-invasive way to improve patient care, and the authors have effectively addressed this important area of recovery.

Reviewer #3: This study is a compelling demonstration of how music can ease preoperative anxiety and postoperative pain in gynecological surgery patients. The analysis is clear and well-executed, with meaningful results that highlight the power of a simple, non-invasive intervention. I appreciate the practical recommendation for nurses to incorporate music, offering a compassionate and effective way to enhance patient care. It’s a thoughtful and impactful contribution to perioperative medicine.

**Do you want your identity to be public for this peer review?** For information about this choice, including consent withdrawal, please see our Privacy Policy

Reviewer #2: **Yes: ** Akash Kumar

Reviewer #3: No

---

## [Editor Report · Acceptance letter]

PONE-D-24-22771R3

PLOS ONE

Dear Dr. Lin,

I'm pleased to inform you that your manuscript has been deemed suitable for publication in PLOS ONE. Congratulations! Your manuscript is now being handed over to our production team.

Kind regards,

on behalf of

Dr. Ahmed Mustafa Rashid

Academic Editor

PLOS ONE